# Gas Chromatographic Fingerprint Analysis for the Comparison of Seized Cannabis Samples

**DOI:** 10.3390/molecules26216643

**Published:** 2021-11-02

**Authors:** Amorn Slosse, Filip Van Durme, Nele Samyn, Debby Mangelings, Yvan Vander Heyden

**Affiliations:** 1Drugs and Toxicology Department, National Institute for Criminalistics and Criminology (NICC), Vilvoordsesteenweg 100, B-1120 Brussels, Belgium; amorn.slosse@just.fgov.be (A.S.); Filip.VanDurme@just.fgov.be (F.V.D.); Nele.samyn@just.fgov.be (N.S.); 2Department of Analytical Chemistry, Applied Chemometrics and Molecular Modelling, Vrije Universiteit Brussel (VUB), Laarbeeklaan 103, B-1090 Brussels, Belgium; Debby.Mangelings@vub.be

**Keywords:** chromatographic fingerprint, alignment optimization, design of experiments, data pre-processing, comparison intra- and inter-location samples

## Abstract

*Cannabis sativa* L. is widely used as recreational illegal drugs. Illicit Cannabis profiling, comparing seized samples, is challenging due to natural Cannabis heterogeneity. The aim of this study was to use GC–FID and GC–MS herbal fingerprints for intra (within)- and inter (between)-location variability evaluation. This study focused on finding an acceptable threshold to link seized samples. Through Pearson correlation-coefficient calculations between intra-location samples, ‘linked’ thresholds were derived using 95% and 99% confidence limits. False negative (FN) and false positive (FP) error rate calculations, aiming at obtaining the lowest possible FP value, were performed for different data pre-treatments. Fingerprint-alignment parameters were optimized using Automated Correlation-Optimized Warping (ACOW) or Design of Experiments (DoE), which presented similar results. Hence, ACOW data, as reference, showed 54% and 65% FP values (95 and 99% confidence, respectively). An additional fourth root normalization pre-treatment provided the best results for both the GC–FID and GC–MS datasets. For GC–FID, which showed the best improved FP error rate, 54 and 65% FP for the reference data decreased to 24 and 32%, respectively, after fourth root transformation. Cross-validation showed FP values similar as the entire calibration set, indicating the representativeness of the thresholds. A noteworthy improvement in discrimination between seized Cannabis samples could be concluded.

## 1. Introduction

The consumption and production of the Cannabis plant (*Cannabis sativa* L.) is widespread throughout the world [1,2]. During the last decade, interest in this psychoactive plant increased due to the legalization of recreational Cannabis use in some countries [3,4]. Furthermore, in the health care sector, research is rapidly evolving with regard to the medicinal properties of Cannabis [5,6].

However, in many countries, Cannabis is still prohibited [7]. In Belgium, this psychodysleptic drug is mainly cultivated indoors and is commonly seized by law enforcement. Yet, despite the high number of seizures in this country, i.e., about 29,000 marijuana confiscations in 2018, and the fact that Cannabis is the most commonly used drug worldwide, research is still lacking with regard to the direct comparison of seized marijuana samples, i.e., Cannabis profiling, for forensic and judicial purposes [8].

Illicit drug profiling can generally be described as the use of drug profiles, containing their physical and chemical characteristics, with the aim to link illegal drug seizures. It covers multiple purposes, such as the determination of the geographical origin, identifying dealer-user networks and gathering information about drug trafficking and networks [9,10,11]. In this study, Cannabis profiling was performed by studying the chemical properties of the crop. This technique may effectively support the information already obtained by law enforcement, thus presenting additional evidence in court [12]. In other words, it may assist in confirming links between suspects, which indicates that seized samples coming from a consumer/dealer all have a common origin.

In the literature, cocaine, amphetamine, and heroin profiling have already been studied [13,14,15]. Subsequently, it has been demonstrated that target-based profiles, consisting of natural components with frequently occurring cutting agents and adulterants, contribute to the successful outcomes of cases in which different seizures of the abovementioned drugs could be reliably linked [13,14,15]. However, Cannabis is an agricultural crop, and like other plants, its intrinsic composition is very complex and highly depends on many growth factors, resulting in an inherent variability in analyte composition between plants coming from the same cultivation [2,16]. This makes it more challenging to develop a method to establish links between seizures.

Moreover, *Cannabis sativa* L. is chemically characterized by over 500 entities containing terpenes, flavonoids, alkaloids, and a unique terpenophenolic group named cannabinoids [12,17,18]. Previously published forensic work mainly focused on data where only specific compounds were taken into account. The most widely studied compounds are the three main cannabinoids Δ^9^-tetrahydrocannabinol (THC) (i.e., the primary psychoactive ingredient), cannabidiol (CBD)m and cannabinol (CBN) (the degradation product of THC) [19,20,21]. Hence, when analysis is restricted to only a limited number of components, it is most probable that the comprehensiveness of the plant material cannot be fully assessed [22]. Therefore, another approach for the identification and subsequent comparison of Cannabis, such as chromatographic fingerprinting, should be taken into consideration.

A herbal/chromatographic fingerprint can be described as a chromatogram in which both major and minor secondary metabolites, many of them unknown and present in small amounts, are revealed. It provides as much information as possible derived from an extract, and it optimally separates all peaks in the chromatogram [23,24]. This approach has already been successfully utilized in several analytical applications, e.g., in the quality control of traditional herbal medicines where the chromatographic fingerprint represents the chemical integrity of the botanical products used for the identification and authentication of herbal drugs and in drug discovery where new components can be discovered with the aim to overcome certain problems (such as drug resistances) with currently applied medicines [25,26,27].

The concept of ‘Cannabis fingerprinting’ has already been mentioned in some research papers. Fischedick et al. (2010) [28] demonstrated that the metabolic fingerprinting of Cannabis, subjected to chemometric techniques, was able to chemotaxonomically differentiate Cannabis varieties. The authors were interested in finding the most discriminating components of the plant when applying principal component analysis. El Sohly et al. (2007) [18] reported a study where entire marijuana chemical fingerprints were used to develop a database with profiles of known origin that could then be utilized to determine the geographical region of origin of other samples. However, to the author’s knowledge, no prior investigation into the use of chromatographic fingerprints as an untargeted approach for illicit drug profiling has been conducted.

Recently, we applied Cannabis profiling by demonstrating the use of a GC–MS chemical profile consisting of selected cannabinoids (multivariate peak table) and multivariate data analysis with the aim to determine a decision-making threshold to link seized Cannabis samples. The developed method used Pearson correlation coefficients after data pre-processing. It was shown that the herbal material could be meaningfully compared, resulting in a significant improvement in discriminating different samples [29]. Based on this study, the current manuscript proposes a chromatographic fingerprint approach in which the complete profile is taken into account as a new method for the direct comparison of seized Cannabis samples coming from different plantations. The same methodology as applied above on the selected peaks can be carried out on entire GC–MS and GC–FID fingerprints to evaluate and compare the intra (within) and inter (between) plantation variabilities in order to distinguish herbal materials.

## 2. Results and Discussion

### 2.1. Similarity of the Cannabis Fingerprints

Each Cannabis sample, originating from ten plantations, was extracted and subjected to chromatographic analysis. A GC–FID chromatographic fingerprint and a total ion chromatogram (TIC), provided by GC–MS, were obtained. Both herbal fingerprints were dominated by a high THC peak. For the GC–MS chromatograms, THC exceeded the detector upper limit and the loading capacity with the used column dimensions and type. Successful cocaine profiling without its main compound was already demonstrated by Broséus et al. [30].

Consequently, the THC peak was rejected from the GC–MS dataset, resulting in 5144 data points instead of 5231. Both Cannabis chromatograms are shown in Figure 1A,B.

First, the samples were quantitatively tested using the accredited routine GC–FID method at the NICC. This was done to study the potency of the indoor cultivated plants and to see whether the total content of THC, i.e., the combined amount of THC and its acidic form, exceeded the legal concentration of 0.2 *m*/*m*% [31]. The total THC levels of the ten cultivation sites ranged from 8 to 21 *m*/*m*%, with CBD and CBN contents below 1 m/m%. This latter component is the degradation product of THC and indicates the freshness of the samples due to its low amount.

It is known that Cannabis is highly inconsistent in its secondary metabolite composition. Sexton and Ziskind [16] even pointed out that Cannabis crops from the same cultivation room show variability in their components. Therefore, it was important to evaluate the similarity of the chromatographic fingerprints per plantation prior to further data analysis. This was done by aligning the datasets and calculating the Pearson correlation coefficients (r). At first, triplicate extractions and injections were executed to study the variability of the used analytical methods. Average r-values of 0.997 and 0.999 were found for GC–MS and GC–FID, respectively. These very high correlations indicate the repeatability of the GC analyses. Subsequently, the intra-correlation coefficients, representing the correlation between two entire profiles from the same cultivation site, were determined. The results were visualized using colour maps showing the correlation coefficient matrices of each plantation. The same intra-correlation outcome was observed for both GC–MS and GC–FID data. For instance, as can be seen in Figure 2A, where the colour map from the GC–FID data is shown, three outlying samples could be differentiated in the fourth plantation. They show lower r-values, i.e., less similarity to the other samples, which had high correlation coefficients (red-brown coloured). Differences in composition could also be observed when looking at the herbal fingerprints of this particular cultivation site (see Figure 2B). Hence, it was decided to remove these samples from the dataset because the differences could lead to unnecessary variability within cultivation sites. No outliers were observed in the other plantations. Moreover, as was also reported in the literature, some other colour maps show naturally occurring heterogeneity (See Appendix A). This was important to take into account for the development of the discriminating model. Consequently, the GC–MS and GC–FID datasets were reduced to 97 samples, with seven remaining Cannabis plants in the fourth plantation. A total of 426 intra and 4230 inter cultivation site r-values were used for all further calculations in this paper.

### 2.2. Pre-Processing of the Raw Data

#### 2.2.1. Alignment Optimization

As can be observed in Appendix A, the GC–FID fingerprints showed some shifts in retention time that were far less for the GC–MS TIC’s. Consequently, aligning both raw datasets by COW was carried out. Warping was executed in order to properly extract the relevant differences between the samples. This first pre-processing step was performed to prepare the dataset for further pre-treatment analysis with the aim of finding an approach that differentiated the seized Cannabis plants as much as possible.

Two different warping techniques, i.e., the fully ACOW technique and the DoE-COW approach, were used to align the raw data. By doing so, the two crucial parameters—defined as the SL and SS—were optimized in two SL ranges of 15–100 and 25–200 and one SS range of 1–10 to select the optimal SL–SS combination. To compare both approaches, the methodology was used as described in Section 3.5.4 to calculate the FNs and FPs. The determined FP error rate was used to evaluate the alignment. Hence, while selecting the best warping technique, the authors of this study also explored whether an improvement of the FPs was seen when aligning the raw data more than once. In Appendix A, the FN and FP error rates for the 95% and 99% CLs are summarized for both COW approaches on the GC–FID dataset. The initial FPs of the raw data were 34% and 39% for both CLs. When aligning the fingerprints once, using ACOW and DoE, a higher FP error rate was obtained. It should be mentioned that the lower FP values of the raw dataset resulted in an incorrectly better discrimination between the different plantations because of the retention time shifts. Therefore, alignment was performed to exclude this chromatographic influence that was causing, as expected, higher similarities between the 10 cultivation sites. Then, for the first SL range, for example, an improvement to 47% FPs at the 95% CL was seen when performing ACOW twice and DoE once. Similar error rate values were obtained for the second SL range when executing DoE three times (best results marked in bold). However, when looking at the overall results, we found no large difference between the two COW approaches.

It was already mentioned that for the GC–MS raw data, only very small retention time variations were observed. This may be explained by the GC–MS column dimensions. A shorter column with a smaller diameter and a thinner liquid film is less susceptible to retention time shifts throughout analyses than the column used for GC–FID. Therefore, as can be seen in Appendix A, alignment optimization had hardly any effect on the raw data with either ACOW or DoE. For the further evaluation of the extra pre-processing methods, ACOW was used to align the GC–MS and GC–FID raw data because this is a fully automated technique for use in lab practice, resulting in the most efficient warping.

#### 2.2.2. Data Treatment—Discrimination Linked and Unlinked Cannabis Samples

Once the alignment was studied and evaluated, statistical analyses were performed to find a pre-processing method that allowed for the best correlation between linked samples from the same cultivation site and the best distinction of unlinked Cannabis samples originating from different plantations. Additional pre-processing was conducted to reduce the influence of larger peaks present in the fingerprint. The pre-treatment methods that were tested on the aligned data were column centering (CC), normalization (N) combined with CC, standard normal variate (SNV) followed by CC, square root/fourth root normalization, and auto-scaling. The first three pre-processing are often applied with chromatographic fingerprints [32,33]. Square root and fourth root normalization are pre-treatment methods often applied with Pearson correlation calculations in illicit drug profiling [13,34]. Auto-scaling was used because it creates equally import chromatographic peaks. The aim of conducting these pre-treatments was to minimize the intra-location and inter-location distributions overlap resulting in an improvement of the FP values. 

At first, a reference dataset was selected. Initially, multiple alignments of the GC–FID fingerprints by ACOW positively affected the FP error rate. The results after twice ACOW plus the additional pre-treatments are shown in Appendix A. However, when comparing these data with the left side of Table 1, the FP values were equivalent to those obtained after a one time ACOW performance. Moreover, after applying fourth root normalization and auto-scaling, an equal decrease in FPs was observed for both aligned GC–FID datasets. Consequently, for the GC–MS and GC–FID fingerprints, it was eventually decided to use the results of the once by ACOW-aligned chromatograms as the reference dataset. For GC–FID, the first SL–SS range was applied, and for GC–MS, the second SL–SS range, i.e., SL 25–200 with SS 1–10, was chosen for further computations. Next, each pre-treatment was evaluated by determining the total % FNs and FPs.

In Table 1, a summary of all pre-processing methods is shown for the GC–FID and the GC–MS data matrices. First, threshold values of 0.989 and 0.987 for the 95% and 99% CLs, respectively, were obtained with no additional pre-treatment for the GC–FID-aligned profiles. For the 95% CL, an FN error rate of 6% and an FP value of 57% were found. The 99% limit showed 4% FNs and 65% FPs. This latter value was very high, meaning that a high number of inter-plantation correlation coefficients were situated above the ‘linked’ threshold and were seen as equal. Such a high FP value was also observed for the GC–MS-warped data. Here, the aforementioned CLs generated thresholds of 0.966 and 0.959, respectively. The obtained FPs were 54% and 57% for the 95% and 99% CLs with 6% and 2% FNs, respectively.

The first three data pre-treatment methods, i.e., CC, N and CC, SNV and CC, did not show much improvement in getting lower FPs for the GC–MS and GC–FID fingerprints. Furthermore, by applying these pre-processing steps, very low or even negative thresholds were obtained. Thresholds were calculated using intra-plantation r-values. Therefore, it was our intention to obtain high correlations, i.e., aiming at values close to one, resulting in very strong positive relations between the samples within cultivation sites. After performing the abovementioned pre-treatments, there was a loss of correlation between the linked samples. This situation also occurred with the use of auto-scaling for the GC–FID dataset, even though the lowest FPs were obtained with this pre-treatment. Consequently, it was decided not to use this pre-processing for further computations. When applying fourth root normalization on the GC–FID-aligned data, the FPs were largely reduced from 57 to 24% and from 65 to 32% FPs for the 95% and 99% CLs, respectively. The FNs were 6 and 4% for both thresholds, with values of 0.988 and 0.985, respectively. A similar outcome was obtained when analysing the GC–MS chromatograms. Here, a noteworthy improvement of FPs was observed from 54 to 28% for the 95% CL, while for the 99% CL, the FP error rate was decreased from 57% to 35%, with 6 and 2% FNs at both CLs. Threshold values of 0.979 and 0.975 for the same CLs, respectively, were obtained. When comparing the results of this best transformation technique for both analytical datasets, it was found that GC–FID showed the highest decrease in FPs, i.e., 24%, compared to 28% for GC–MS. A possible reason for the difference could be that GC–FID has a larger linear range with less overlapping peaks because of the detector properties and the used GC column. Therefore, it could be concluded that by using GC–FID, a better distinction was obtained between the herbal Cannabis fingerprints. This dataset was used for further elaboration in this study.

The effect of the fourth root normalization is emphasized in Figure 3. This figure illustrates the frequencies of the r-values between the intra- and inter-location samples for both the reference dataset and after fourth root normalization. Figure 3A,B shows that a considerable decrease was obtained between the intra- and inter-location distributions overlap after the pre-treatment. It can be stated that this pre-treatment was best capable of differentiating between the ten plantations.

The FP improvement could also be illustrated by ROC curves (Figure 4), which show that the fourth root normalization curve moved faster to the upper left corner of the curve to approach a sensitivity value of 1, resulting in a better discrimination between the Cannabis samples. To measure the performance of the methods, AUCs were calculated for the reference data and the best pre-treatment; Table 2 presents these AUCs, which were 0.834 and 0.947, respectively. Therefore, it can be concluded that after using fourth root normalization on the aligned data, a significant decrease in FPs was generated, thus obtaining a higher AUC. Additionally, this was in agreement with our previous paper [29], where the fourth root normalization also acquired the best results to distinguish plantations.

### 2.3. THC Influence

Δ^9^-THC is widely known as the main constituent of Cannabis that causes the mind-altering effect of the drug. Consequently, in the GC fingerprints, this compound appeared as the major peak in the chromatograms. For the GC–MS TICs, THC was excluded for further investigation because of detector limitations for certain component ions. Since GC–FID is the analytical technique that provides the greatest improvement of the FP error rate, the authors of this paper also wanted to study the influence of THC on these Cannabis fingerprints. Hence, after the removal of the component of the ACOW-aligned data, resulting in a data matrix consisting of 8037 time points, all above-discussed pre-processing methods were applied (Appendix A). After fourth root normalization, which was the best pre-treatment, FP values of 28 and 37% for the 95% and 99% CLs, respectively, were obtained. Comparing these results with the FPs of the data containing THC—i.e., 24 and 32%, respectively—revealed a negative effect on the overall results after removing the main constituent data from the GC–FID fingerprints.

### 2.4. Cross-Validation

After transforming the aligned GC–FID data using fourth root normalization, it was necessary to evaluate the predictive performance of this pre-treatment method. Therefore, cross-validation (CV), subdividing the data matrix into test sets and calibration sets, was used to study the effectiveness of the obtained thresholds. The two approaches that were applied were leave-one-plantation-out CV (LOPO-CV) and leave-*n*-objects-out CV (LNO-CV). In a previous paper [29], these approaches were extensively discussed. Briefly, the first approach uses a test set containing 5% of the intra- and inter-location r-values. The remaining correlation values for the linked samples were used to determine the 95% and 99% CL thresholds. This was repeated 20 times in order to process the entire dataset, and the overall FN% and FP% were computed.

For LOPO-CV, one plantation at a time was used as a test set. This approach was also investigated because here, information can be gathered about particular cultivation sites. The 95% and 99% CLs were derived from the intra-plantation r-values of the nine remaining plantations. This cross-validation method was performed ten times to eventually calculate the overall FNs and FPs. Table 3 summarizes the accuracy of the approaches, shown as FNs and FPs, in predicting the test sets. Both CV approaches showed similar results. LNO-CV had FN rates of 6% and 4% for the 95% and 99% CLs. The acquired FP rates for both CLs were 24 and 32%, respectively. For the LOPO-CV approach, FNs of 7 and 4% for the 95% and 99% CLs were derived with 25 and 33% FPs, respectively. Subsequently, the predictive ability of the CV was compared with the results obtained when applying the fourth root normalization on the entire 97 GC–FID fingerprints dataset (see Table 3). A similar output was found. This demonstrates that the computed ‘linked’ thresholds were representative for the overall GC–FID data matrix.

## 3. Materials and Methods

### 3.1. Origin of Samples

All Cannabis plants were collected by members of the Belgian Police during house/building searches where illicit cultivations were discovered. In total, 100 mature female plants, coming from 10 different regions in Belgium (i.e., 10 plants per cultivation site), were seized and analysed. The cultivation sites are numbered from 1 to 10 further in the text.

### 3.2. Sample Preparation

The flowering buds of each plant were collected and dried at 40 °C for at least 12 h and subsequently ground in an electric mill (IKA^®^ Tube Mill Control, Staufer, Germany). Then, 100 ± 10 mg of dry homogenized herbal Cannabis was automatically extracted using the Tecan Freedom Evo 150 liquid handling platform (Tecan Trading, Männedorf, Switzerland). The extraction procedure was carried out by adding 10.0 mL of 99% denaturated ethanol (VWR BDH Prolabo Chemicals, Leuven, Belgium)—containing 0.01 mg/mL of internal standard, tribenzylamine (Alfa Aesar, Karlsrühe, Germany)—to each sample. Subsequently, the Tecan^®^ instrument sonicated all samples for 15 min, followed by 5 min of horizontal shaking (100 turns). Afterwards, 1.0 mL of the obtained extracts was collected twice in 1 mL vials and subjected to one of the chromatographic methods.

### 3.3. GC–MS Conditions

An Agilent^®^ 6890N gas chromatograph (Agilent Technologies, Santa Clara, CA, USA) coupled to an Agilent^®^ 5973N mass selective detector was used to perform the analysis of the seized Cannabis samples. The system was provided with a DB5-ms (5% diphenyl and 95% dimethyl silica) capillary column (15 m × 0.25 mm i.d., 0.25 µm film thickness; J&W Scientific, Folsom, CA, USA). The injection volume was set at 2 μL. Helium was used as the carrier gas at a flow rate of 1.3 mL/min and a split of 1:10. The injection temperature was 230 °C. The GC oven program was initiated at 60 °C, with subsequent increases of 8.5 °C/min to 240 °C, where it was held for 1.75 min. Finally, at a 30 °C/min rate, the temperature was raised to 320 °C and held for 3 min. The total run time was about 29 min. The mass analyser operated by electron impact (70 eV). The ion source and single quadrupole analyser were set at 300 and 150 °C, respectively. Full spectra were obtained over a 50–500 amu mass range.

### 3.4. GC–FID Conditions

Chromatographic analyses were carried out with an Agilent^®^ GC–FID 7980B system (Agilent Technologies) equipped with a flame ionization detector (FID). A HP-5 ((5%-phenyl)-methylpolysiloxane-phase) column (25 m × 0.32 mm i.d. × 0.52 μm film thickness; J&W Scientific, Folsom, CA, USA) was used. The flow rate of the carrier gas, i.e., helium, was set at 2.56 mL/min, with an H_2_ flow of 30 mL/min and an airflow rate of 400 mL/min for the FID. N_2_ was used as make up gas, with a flow rate of 25 mL/min. The injections were performed by an Agilent 7693 Series injector. A 2 μL sample was injected with a split ratio of 1:10. The detector and injection temperatures were maintained at 320 and 230 °C, respectively. The temperature program was the following: the oven temperature was initially set at 60 °C, then increased at 8.5 °C/min to 280 °C and held for 5 min, and finally increased to 320 °C at a rate of 20 °C/min. The total time for each GC run was about 34 min.

### 3.5. Data Analysis

All chromatographic fingerprints were generated as text files from the Enhanced Data Analysis program of MSD Chemstation E.02.01.1177 (GC–MS) (Agilent Technologies) and of CDS Chemstation A.02.05.021 (GC–FID) (Agilent Technologies). To import the fingerprint files in the Matlab^®^ software and finally compute the false negative (FN) and false positive (FP) errors, Microsoft^®^ Excel 2013 (Microsoft Corporation, Redmond, WA) was used. Further data acquisition and data (pre-)processing such as the alignment optimization, the Pearson correlation coefficient calculations, and threshold determinations were carried out with Matlab™ R2020a (The Mathworks, MA, USA).

#### 3.5.1. Alignment Optimization

A first data matrix originating from GC–MS consisted of 100 samples, each with 5231 time points at which the signal intensity was measured. GC–FID analyses were performed on the same samples, resulting in a second data matrix consisting of 100 samples × 8101 variables. In chromatography, small unavoidable experimental variations between runs occur. This causes minor retention time variations that complicate chemometric analysis. Consequently, to apply multivariate data analysis, it is important that the variability between samples is reflected as chemical differences between fingerprints, though these should not be affected by experimental errors and cause retention time variability [35]. In order to correctly compare the data, peak alignment is a necessary step.

##### Automated Correlation-Optimized Warping (ACOW)

Correlation-optimized warping (COW) is a classic chromatographic pre-treatment technique that corrects retention-time differences by aligning the GC profiles. This procedure consists of the following steps. At first, a reference chromatogram (or target chromatogram) is automatically selected from the dataset. Subsequently, the sample chromatogram to align and the target chromatogram are divided in segments, which are specified by the segment length (SL). The correction is then conducted by warping these segments relatively to those of the reference profile. Hence, the linear stretching and compression of the length are carried out by shifting the end points with a limited number of data points, defined as the slack size (SS) [36]. The aim of this technique is to find the optimal correction that results in a maximum correlation with respect to the target fingerprint. To achieve this goal, the two warping parameters, i.e., the segment length and the slack size, need to be optimized because they have significant influence on the results [37,38]. Skov et al. (2006) [39] developed an automated method that corrects the shifts in retention time while preserving the peak shapes and areas in the chromatogram. The optimization of the alignment parameters is based on a discrete-coordinate simplex algorithm. To select the most satisfying SL and SS for the overall dataset, the method performs a global space search for each parameter. After correspondence with the authors, it was decided that for the Cannabis fingerprints, two SL ranges of 15–100 and 25–200 and one SS range of 1–10 would be evaluated in this study. After the determination of the best parameters, the raw fingerprint data were aligned with the obtained SS and SL combination. For more details on the applied algorithms, we refer to [39,40,41].

##### Design of Experiments (DoE)

Alternatively, a design of experiment (DoE) approach was set up to determine the optimum SL–SS combination for the GC fingerprints. We performed full factorial 3^2^ design consisting of two factors (SL and SS), each at three levels, i.e., low (−1), intermediate (0), and high (1), as shown in Figure 5. Each point in the figure represents the factor levels for one experiment. The same SL and SS ranges described in previous section were also used here when applying the two designs. Table 4 presents the factor levels for the experimental designs. The model for the response is described as:*y* = *β*_0_ + *β*_1_ + *β*_2_*x*_2_ + *β*_11_*x*_1_^2^ + *β*_22_*x*_2_^2^ + *β*_12_*x*_1_*x*_2_(1)
where *y* is the response, *x*_1_ and *x*_2_ are the factors, *β*_0_ is the intercept, *β*_1_ and *β*_2_ are the linear factor coefficients, *β*_11_ and *β*_22_ are the quadratic coefficients, and *β*_12_ is the coefficient representing the two-factor interaction. As a response, the average of all intra-plantation Pearson correlation coefficients in the data matrix was used. Consequently, each dataset was optimized by performing nine experiments. The DoE experiment showing the highest average intra-plantation correlations was selected to provide the optimal SL and SS pair to warp the initial raw data profiles. More information about the use of DoE can be found in [42,43,44].

#### 3.5.2. Data Pre-Processing Techniques

After alignment, other pre-treatment techniques on the chromatographic data were carried out. This was done to reduce experimental error, decrease the large concentration differences of the predominant components, and enhance the discrimination between the Cannabis samples from different cultivation sites. The following procedures were applied: column centring (CC), normalization (N) followed by CC, standard normal variate (SNV) combined with CC, square root/fourth root transformation [14,45], and auto-scaling. Column centring (CC) computes a mean value for each column of the warped data matrix and subtracts this respective mean from each column element. Normalization scales the fingerprints to a constant total by dividing each row element by its row standard deviation, defined as row scaling. SNV pre-processing combines row centring, i.e., each respective row mean is subtracted from each row element, and row scaling [46]. Auto-scaling refers to column centring followed by dividing by the column standard deviation [47]. For more background information on the applied methods, we refer to [48,49,50]. The importance and effect of applying data pre-processing techniques to compare seized drugs were already investigated in earlier illicit drug profiling research [30,34,45].

#### 3.5.3. Pearson Correlation Coefficient

The Pearson correlation coefficient parameter, which quantifies the linear relationship between two series of measurements, was already used in different illicit drug profiling studies aiming to discriminate between linked and unlinked seized samples [13,14]. As was more comprehensively described in earlier work [29], here, the Pearson correlation coefficient (PCC), also denoted as ‘r’, provided a pairwise comparison of the herbal fingerprints. When calculating r, the degree of similarity between two chromatograms is assessed. These coefficients range from −1 to +1. The value +1 indicates a perfect similarity with a positive linear correlation, whereas −1 represents a perfect negative correlation [51]. The latter is irrelevant when comparing fingerprints. In this study, r was used as similarity parameter to compare the seized Cannabis samples within and between the ten cultivation sites.

#### 3.5.4. False Negative and False Positive Error Rates in Forensic Science

Experimental testing in science inevitably has to deal with the occurrence of some errors leading to false negatives (FNs) and false positives (FPs). This also applies to the forensic field [52,53]. Previous research has already shown that even after using different data pre-processing methods to compare seized samples, an overlap of the obtained sample distributions is seen, resulting in a certain percentage of erroneous results defined as FNs and FPs [29]. These FN values are obtained when samples are seen as different when they, in fact, originate from the same source. The opposite is defined as FPs, i.e., situations where the forensic expert concludes that the samples match but, in fact, come from different sources. Considering these error rates is important to properly interpret findings. Furthermore, these values are crucial to indicate the reliability of the used method. For illicit drug profiling, where the results may be admitted as additional evidence in court, there is a preference to minimize FPs. Consequently, it can then be concluded with a high degree of certainty whether seized samples are related [53]. In order to show a link between samples, a forensic expert needs to determine a threshold value as decision-making parameter. This is accomplished by combining statistical methodology, i.e., confidence limits (CLs), with similarity analysis [45].

In this paper, where Gaussian-distributed data were assumed, two frequently used confidence levels, i.e., 95% and 99% CLs, were determined [54].

##### Discriminating Power Methodology

As proposed in [29], the following methodology was applied to evaluate the intra- and inter-location variability and to determine the discriminating ability after each pre-processing approach. A schematic overview is presented in Figure 6. The obtained datasets for both chromatographic techniques consisted of intra- and inter-location samples. Intra-location samples, i.e., ‘linked’ samples, are samples coming from the same cultivation site, whereas inter-location samples, i.e., ‘unlinked’ samples, are samples from different plantations. At first, r was calculated for each sample pair in the data matrix. Then, to derive the threshold values of the abovementioned 95% and 99% CLs, the correlation coefficients within each plantation, i.e., intra-location r, were used. These r-values were seen as ‘linked’ correlations because they all came from the same source, theoretically resulting in high correlations. Consequently, when a correlation was found between two samples above the threshold, it could be concluded that these samples were related. For normally distributed data, 95% of the intra-location r-values were situated above the limit $\bar{r}$ − 1.96s_r_, where $\bar{r}$ is the average of the intra-location correlation coefficients of the linked samples and s_r_ is its respective standard deviation. We found that 99% of the r-values were located above the limit $\bar{r}$ − 2.576s_r_. This methodology allowed a known error rate, i.e., 5% and 1% FNs. Next, to determine the degree of distinction between all cultivation sites, the FP error rate was computed by using the correlation coefficients of unlinked samples, i.e., the r-values calculated between inter-location samples. The aim of this paper was to find the best pre-treatment resulting in an acceptable FP value so that the found threshold could be used to link seized Cannabis samples. Finally, after the selection of the best pre-processing technique with the best FP response, the predictive performance was evaluated using cross-validation. This was done via two different approaches, i.e., leave-one-plantation-out (LOPO) cross-validation and leave-*n*-objects-out (LNO) cross-validation (CV).

##### ROC Analysis

Drawing a receiver operating characteristics (ROC) graph is a common approach to visualize the overall performance of a method. This plot represents the sensitivity, defined as the true positive values, as a function of 1-specificity, i.e., the FP error rate. Each curve is a collection of sensitivity–specificity combinations for a number of calculated thresholds. Here, to measure the accuracy of a pre-processing method to differentiate between the cultivation sites, the area under the ROC curve (AUC) was used. The closer the AUC was to 1, the more the pre-treatment improved the distinction of the samples, causing less FPs. The approach also uses an AUC value of 0.5, which represents the total overlap of the intra- and inter-location distributions, resulting in zero discrimination [29,55].

## 4. Conclusions

This paper has demonstrated the use of a rather large dataset consisting of 97 GC–MS or GC–FID chromatographic fingerprints for Cannabis profiling purposes. The entire pattern of the compounds, not a selection of compound peaks, was analysed. The study emphasized the application of different pre-treatment methods during data analysis to discriminate between different cultivation sites. ‘Linked’ fingerprint thresholds, i.e., the 95% and 99% CLs, were derived from the r-values of the intra-location data. To evaluate the effect of the pre-processing, FNs and FPs were determined with the aim of minimizing the FP error rate, which resulted in a decreased overlap between the correlation coefficient distributions and an improved differentiation between the seized Cannabis samples. First, the alignment optimization of the herbal fingerprints was conducted to eliminate the minor retention time shifts between fingerprints. Two different approaches that showed similar results, i.e., ACOW and DoE, were considered. The ACOW dataset was used as the reference dataset for further computations because ACOW is a fully automated approach, resulting in a more efficient data processing. Subsequently, fourth root normalization as data pre-treatment was the most efficient method, with a noteworthy decrease in FPs for both GC datasets. When comparing the fourth root transformation results of the two analytical techniques with the reference data, it was seen that the GC–FID fingerprints acquired the best discrimination between the ten different plantations, with a large decrease in FPs (from 57 to 24% for the 95% CL). Two different cross-validation approaches, i.e., LNO-CV and LOPO-CV, showed similar results as when evaluating the entire calibration set. Therefore, it can be stated that the derived ‘linked’ fingerprint thresholds are representative for estimating the overall within-plantation variation. The aim of the study was to obtain an FP error rate below 10%. Unfortunately, the final FP rate of 24% was too high, but it could be decreased by allowing for a higher number of FNs in order to use this approach as additional evidence in court. In the future, chemical profiles containing specifically selected cannabinoids will be derived from this dataset and treated according to the same approach, followed by a comparison with the results from the entire fingerprints.

## Figures and Tables

**Figure 1 molecules-26-06643-f001:**
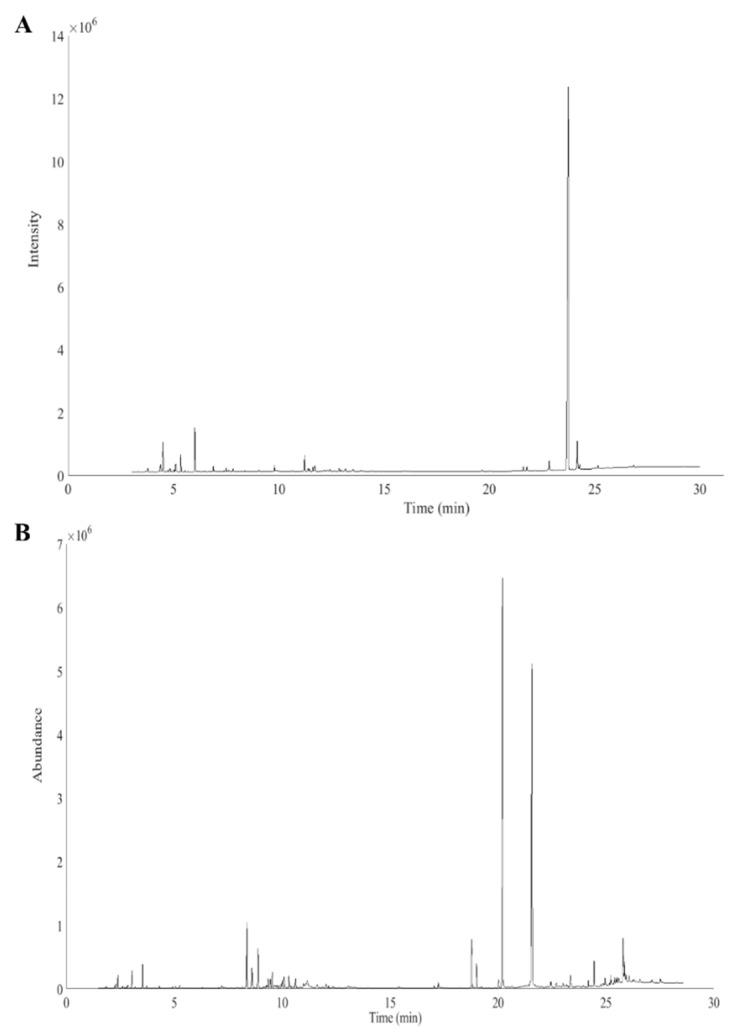
Cannabis fingerprints derived from (**A**) GC–FID and (**B**) GC–MS (without THC peak).

**Figure 2 molecules-26-06643-f002:**
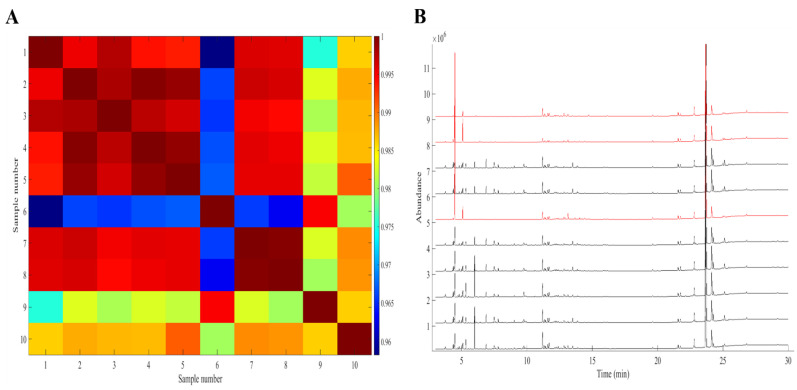
(**A**) GC–FID intra-correlation coefficients colour map showing three outlying samples (6, 9, and 10) in the fourth cultivation site. (**B**) GC–FID fingerprints of the Cannabis samples from the fourth plantation. The outlying chromatograms are plotted in red.

**Figure 3 molecules-26-06643-f003:**
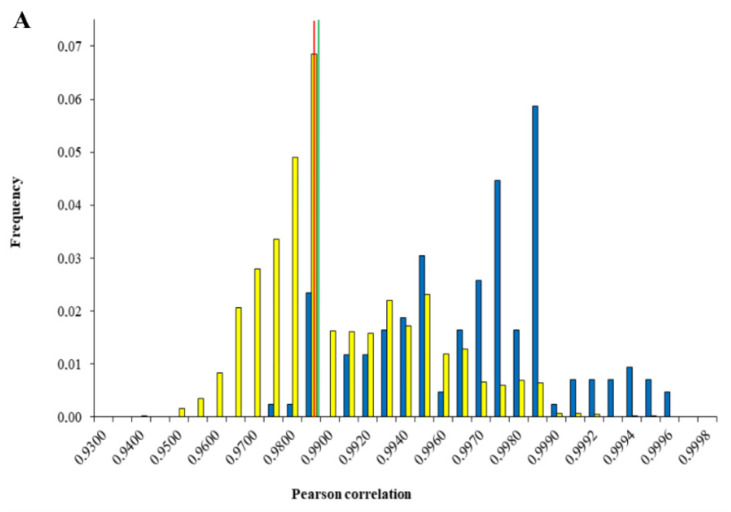
Histograms of the Pearson correlation coefficients and their respective distribution overlap for (**A**) the reference data (GC–FID-aligned data) and (**B**) after fourth root normalization. The yellow bar chart corresponds to the coefficients of the inter-location samples, while the blue bar chart refers to the intra-location correlation coefficients. The two confidence limits are represented as the green vertical line for the 95% CL and the red vertical line for the 99% CL.

**Figure 4 molecules-26-06643-f004:**
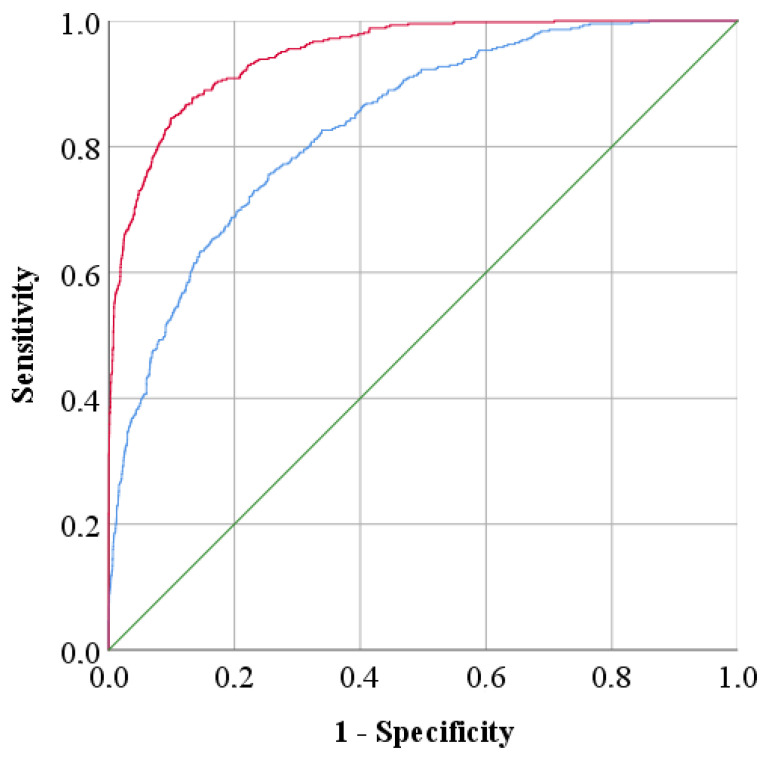
ROC curves representing the accuracy of the reference data (blue line; AUC = 0.834) and after fourth root normalization (red line; AUC = 0.947). The green diagonal line corresponds to the reference AUC = 0.5.

**Figure 5 molecules-26-06643-f005:**
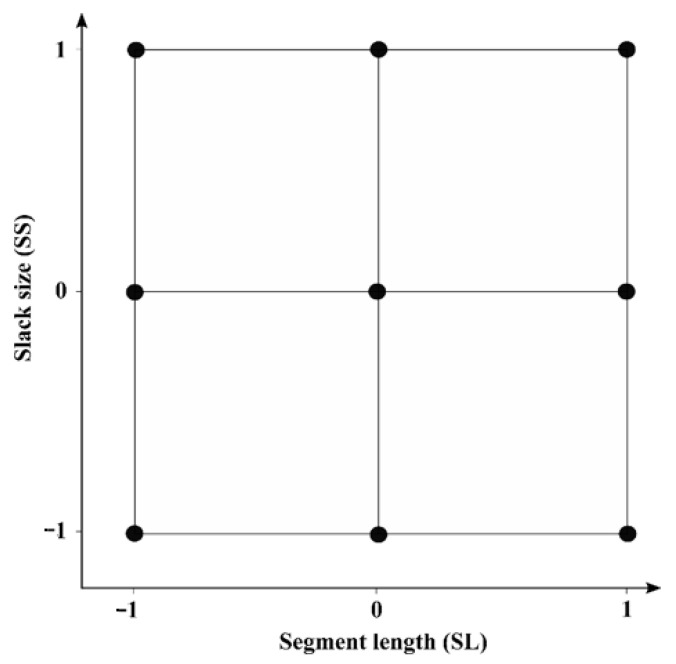
Three-level full factorial design, with −1 representing the lowest level of both factors (SL and SS), 0 representing the intermediate level, and 1 representing the highest level.

**Figure 6 molecules-26-06643-f006:**
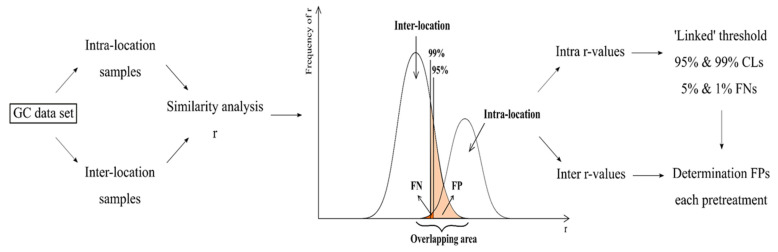
Schematic overview of the methodology used to evaluate the discriminating power of the studied pre-treatments.

**Table 1 molecules-26-06643-t001:** An overview of the total FN% and FP% from all studied pre-treatment methods for both GC–FID and GC–MS after aligning once with ACOW. The fourth root normalization, i.e., the best pre-treatment, is marked in bold.

	GC–FID				GC–MS			
Pre-Treatment Method	95% CL		99% CL		95% CL		99% CL	
	FN (%)	FP (%)	FN (%)	FP (%)	FN (%)	FP (%)	FN (%)	FP (%)
**Aligned data (1×ACOW)**	6	57	4	65	6	54	2	57
**Column centring (CC)**	9	55	5	65	7	51	2	56
**Normalization and CC (N and CC)**	6	57	4	64	5	48	2	53
**Standard normal variate and CC (SNV and CC)**	7	52	4	65	5	48	2	53
**Square root**	6	30	4	39	4	38	2	46
**Fourth root**	**6**	**24**	**4**	**32**	**6**	**28**	**2**	**35**
**Auto-scaling**	6	19	4	27	6	64	0	87

**Table 2 molecules-26-06643-t002:** The generated AUCs and the 95% confidence interval for both the reference data and after fourth root normalization.

Data	AUC	95% Confidence Interval	
		Lower Limit	Upper Limit
**Reference**	0.834	0.815	0.853
**After Fourth Root Normalization**	0.947	0.938	0.957

**Table 3 molecules-26-06643-t003:** The total FN and FP error rates of the CV approaches. The entire calibration dataset consisted of 426 intra-plantation r-values with 4230 inter-plantation correlation coefficients.

	% Misclassifications			
Cross-Validation Approach	95% CI Limit		99% CI Limit	
	FN	FP	FN	FP
**Leave-*n*-Out (LNO)**	6	24	4	32
**Leave-One Plantation-Out (LOPO)**	7	25	4	33
**Entire Calibration Dataset**	6	24	4	32

**Table 4 molecules-26-06643-t004:** SL and SS level values used in the full factorial designs.

Design	Segment Length (SL)			Slack Size (SS)		
	−1	0	1	−1	0	1
**1**	15	58	100	1	6	10
**2**	25	113	200	1	6	10

## Data Availability

The datasets for this study can be obtained from the corresponding author upon reasonable request.

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
