# Peer review of "Gas Chromatographic Fingerprint Analysis for the Comparison of Seized Cannabis Samples"

_molecules, 2021, doi:10.3390/molecules26216643_

Round 1

Reviewer 1 Report

Dear authors, congratulations on your manuscript which reports an interesting application of chemometric analysis to fingerprint profiles.

I consider the manuscript to be publishable in its current version but
only suggest a couple of minor changes.
1)
I recommend trying to improve the quality of figure S1 by zooming in
on the chromatograms in order to make the differences between the samples
more easily evident and moving this figure into the manuscript
in the
manuscript so that it is placed side by side with figure 2
2) I suggest introducing some more details on the algorithms used 
for the calculation

Author Response

Reviewer #1

Point 1: I recommend trying to improve the quality of figure S1 by zooming in on the chromatograms in order to make the differences between the samples more easily evident and moving this figure into the manuscript in the manuscript so that it is placed side by side with figure 2     

Response 1: We understand the reviewer’s comment. Consequently, figure S1 was zoomed in resulting in a clearer distinction of the three outliers compared with the other samples. This supplemental figure was implemented in the paper and is denoted as Figure 2B. Figure 2A refers to the respective colour map.

Point 2: I suggest introducing some more details on the algorithms used 
for the calculation

Response 2: Adding details on the algorithms used would, in our opinion, lengthen  the paper considerably, while they are described properly in the literature. Therefore, when discussing the methods, such as ACOW, DoE and the used pretreatments, we additionally refer to the relevant papers for further background information.

Reviewer 2 Report

I have reviewed the manuscript entitled "Gas chromatographic fingerprint analysis for the comparison of seized cannabis samples" by Yvan Vander Heyden, Amorn Slosse, Filip Van Durme, Nele Samyn, and  Debby Mangelings submitted to Molecules as part of the Special Issue Chromatography and Chemometrics 2021, and have the following comments: 

The manuscript is well written.

The materials and methods described in details the methodology, pre-processing and data analysis properly.

The results and discussion are adequate and well written.

I propose this manuscript to be accepted in present form for publication.

Author Response

Reviewer #2

No further modifications/suggestions of the manuscript were necessary.